# On the Embedding Collapse When Scaling up Recommendation Models

## Abstract

Recent advances in deep foundation models have led to a promising trend of developing large recommendation models to leverage vast amounts of available data. However, we experiment to scale up existing recommendation models and observe that the enlarged models do not improve satisfactorily. In this context, we investigate the embedding layers of enlarged models and identify a phenomenon of *embedding collapse*, which ultimately hinders scalability, wherein the embedding matrix tends to reside in a low-dimensional subspace. Through empirical and theoretical analysis, we demonstrate that the feature interaction module specific to recommendation models has a *two-sided effect*. On the one hand, the interaction restricts embedding learning when interacting with collapsed embeddings, exacerbating the collapse issue. On the other hand, feature interaction is crucial in mitigating the fitting of spurious features, thereby improving scalability. Based on this analysis, we propose a simple yet effective *multi-embedding* design incorporating embedding-set-specific interaction modules to capture diverse patterns and reduce collapse. Extensive experiments demonstrate that this proposed design provides consistent scalability for various recommendation models.

## 1 Introduction

Recommender systems are significant machine learning scenarios that predict users' actions on items based on multi-field categorical data (Zhang et al., 2016), which play an indispensable role in our daily lives to help people discover information about their interests and have been adopted in a wide range of online applications, such as E-commerce, social media, news feeds, and music streaming. Recently, researchers have developed deep-learning-based recommendation models to dig feature representations flexibly. These models have been successfully deployed across a multitude of application scenarios, thereby demonstrating their widespread adoption and effectiveness.

In recommender systems, there is a tremendous amount of Internet data, while mainstream models typically tuned with embedding size 10 (Zhu et al., 2022) do not adequately capture the magnitude of the available data. Motivated by the advancement of large foundation models (Kirillov et al., 2023; OpenAI, 2023; Radford et al., 2021; Rombach et al., 2022), which benefit from increasing parameters, it would be a promising trend to scale up the recommendation model size. However, FIX when scaling up the embedding size, the *bottleneck* of mainstream recommendation models (Qu et al., 2016; Lian et al., 2018; Wang et al., 2021), we find an unsatisfactory improvement or even performance drop as shown in Figure 1a. This suggests a deficiency in the scalability of existing architecture designs, constraining the maximum potential for recommender systems.

We take a spectral analysis on the learned embedding matrices based on singular value decomposition and exhibit the normalized singular values in Figure 1b. Surprisingly, most singular values are significantly small, i.e., the learned embedding matrices are nearly low-rank, which we refer to as the *embedding collapse* phenomenon. With the enlarged model size, the model does not learn to capture a larger dimension of information, implying a learning process with ineffective parameter utilization, which restricts the scalability.

In this work, we study the mechanism behind the embedding collapse phenomenon through empirical and theoretical analysis. We shed light on the two-sided effect of the feature interaction module, the characteristic of recommendation models to model higher-order correlations, on model scalability. On the one hand, interaction with collapsed embeddings will constrain the embedding learning

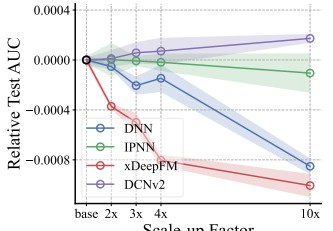  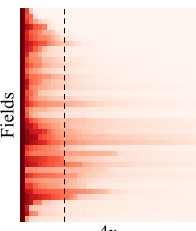 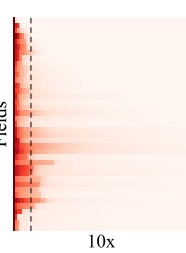

(a) Performance when scaling up recommendation models

(b) Singular values of DCNv2 under different model size, with the dashed lines corresponding to the base size.

Figure 1: Unsatisfactory scalability of existing recommendation models. **(a)**: Increasing the embedding size does not improve remarkably or even hurts the model performance. **(b)**: Most embedding matrices do not learn large singular values and tend to be low-rank.

and, thus, in turn, aggravate the collapse issue. On the other hand, the feature interaction also plays a vital role in reducing overfitting when scaling up models.

Based on our analysis, we conclude the principle to mitigate collapse without suppressing feature interaction about how to design scalable models. We propose *multi-embedding* as a simple yet efficient design for model scaling. Multi-embedding scales the number of independent embedding sets and incorporates embedding-set-specific interaction modules to jointly capture diverse patterns. Our experimental results demonstrate that multi-embedding provides scalability for extensive mainstream models, pointing to a methodology of breaking through the size limit of recommender systems.

Our contributions can be summarized as:

- To the best of our knowledge, we are the first to point out the non-scalability issue for recommendation models and discover the embedding collapse phenomenon, which is an urgent problem to address for model scalability.

- We shed light on the two-sided effect of the feature interaction process on scalability based on the collapse phenomenon using empirical and theoretical analysis. Specifically, feature interaction leads to collapse while providing essential overfitting reduction.

- Following our concluded principle to mitigate collapse without suppressing feature interaction, we propose *multi-embedding* as a simple unified design, which consistently improves scalability for extensive state-of-the-art recommendation models.

## 2 PRELIMINARIES

Recommendation models aim to predict an action based on features from various fields. Throughout this paper, we consider the fundamental scenario of recommender systems, in which categorial features and binary outputs are involved. Formally, suppose there are $N$ fields, with the $i$-th field denoted as $\mathcal{X}_i = \{1, 2, ..., D_i\}$ where $D_i$ denotes the field cardinality. The value of $D_i$ may vary in a wide range, adding difficulty to recommender systems. Let

$$\mathcal{X} = \mathcal{X}_1 \times \mathcal{X}_2 \times ... \times \mathcal{X}_N$$

and $\mathcal{Y} = \{0, 1\}$, then recommendation models aim to learn a mapping from $\mathcal{X}$ to $\mathcal{Y}$. In addition to considering individual features from diverse fields, there have been numerous studies (Koren et al., 2009; Rendle, 2010; Juan et al., 2016; Guo et al., 2017; Lian et al., 2018; Pan et al., 2018; Sun et al., 2021; Wang et al., 2021) within the area of recommender systems to model combined features using *feature interaction* modules. In this work, we investigate the following widely adopted architecture for mainstream models. A model comprises: (1) embedding layers $\boldsymbol{E}_i \in \mathbb{R}^{D_i \times K}$ for each field, with embedding size $K$; (2) an interaction module $I$ responsible for integrating all embeddings into a combined feature scalar or vector; and (3) a subsequent postprocessing module $F$ used for prediction purposes, such as MLP and MoE. The forward pass of such a model is formalized as

$$\begin{aligned} \boldsymbol{e}_i &= \boldsymbol{E}_i^\top \boldsymbol{1}_{x_i}, \ \forall i \in \{1, 2, ..., N\}, \\ h &= I(\boldsymbol{e}_1, \boldsymbol{e}_2, ..., \boldsymbol{e}_n), \\ \hat{y} &= F(h), \end{aligned}$$

where $\mathbf{1}_{x_i}$ indicates the one-hot encoding of $x_i \in \mathcal{X}_i$, in other words, $\boldsymbol{e}_i$ refers to (transposed) $x_i$-th row of the embedding table $\boldsymbol{E}_i$.

## 3 EMBEDDING COLLAPSE

Singular value decompostion has been widely used to measure the collapse phenomenon (Jing et al., 2021). In Figure 1b, we have shown that the learned embedding matrices of recommendation models are approximately low-rank with some extremely small singular values. To determine the degree of collapse for such matrices with low-rank tendencies, we propose *information abundance* as a generalized quantification.

**Definition 1 (Information Abundance)** *Consider a matrix $\boldsymbol{E} \in \mathbb{R}^{D \times K}$ and its singular value decomposition $\boldsymbol{E} = \boldsymbol{U}\boldsymbol{\Sigma}\boldsymbol{V} = \sum\limits_{k=1}^{K} \sigma_k \boldsymbol{u}_k \boldsymbol{v}_k^\top$, then the* information abundance *of $\boldsymbol{E}$ is defined as*

$$\text{IA}(\boldsymbol{E}) = \frac{\|\boldsymbol{\sigma}\|_1}{\|\boldsymbol{\sigma}\|_\infty},$$

*i.e., the sum of all singular values normalized by the maximum singular value.*

Intuitively, a matrix with high information abundance demonstrates a balanced distribution in vector space since it has similar singular values. In contrast, a matrix with low information abundance suggests that the components corresponding to smaller singular values can be compressed without significantly impacting the result. Compared with matrix rank, information abundance can be regarded as a simple extension by noticing that $\text{rank}(\boldsymbol{E}) = \|\boldsymbol{\sigma}\|_0$, yet it is applicable for non-strictly low-rank matrices, especially for fields with $D_i \gg K$ which is possibly of rank $K$. We calculate the information abundance of embedding matrices for the enlarged DCNv2 (Wang et al., 2021) and compare it with that of randomly initialized matrices, shown in Figure 2. It is observed that the information abundance of learned embedding matrices is extremely low, indicating the embedding collapse phenomenon.

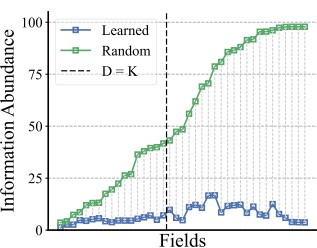

Figure 2: Visualization of information abundance on the Criteo dataset. The fields are sorted by their cardinalities.

## 4 FEATURE INTERACTION REVISITED

In this section, we delve deeper into the embedding collapse phenomenon for recommendation models. Our investigation revolves around two questions: (1) How is embedding collapse caused? (2) How to properly mitigate embedding collapse for scalability? Through empirical and theoretical studies, we shed light on the two-sided effect of the commonly employed feature interaction module on model scalability.

### 4.1 INTERACTION-COLLAPSE THEORY

To determine how feature interaction leads to embedding collapse, it is inadequate to directly analyze the raw embedding matrices since the learned embedding matrix results from interactions with all other fields, making it difficult to isolate the impact of field-pair-level interaction on embedding learning. Under this obstacle, we propose empirical evidences on models with *sub-embeddings* and theoretical analysis on general models, and conclude that feature interaction causes embedding collapse, named the *interaction-collapse theory*.

**Evidence I: Experiments on FFM.** Field-aware factorization machines (FFM) (Juan et al., 2016) split an embedding matrix of field $i$ into multiple sub-embeddings with

$$\boldsymbol{E}_i = \left[ \boldsymbol{E}_i^{\rightarrow 1}, \boldsymbol{E}_i^{\rightarrow 2}, ..., \boldsymbol{E}_i^{\rightarrow (i-1)}, \boldsymbol{E}_i^{\rightarrow (i+1)}, ..., \boldsymbol{E}_i^{\rightarrow N} \right],$$

where sub-embedding $\boldsymbol{E}_i^{\rightarrow j} \in \mathbb{R}^{D_i \times K/(N-1)}$ is only used when interacting field $i$ with field $j$ for $j \neq i$. To determine the collapse of sub-embedding matrices, we calculate $\text{IA}(\boldsymbol{E}_i^{\rightarrow j})$ for all $i, j$ and show them in Figure 3a. For convenience, we pre-sort the field indices by the ascending order

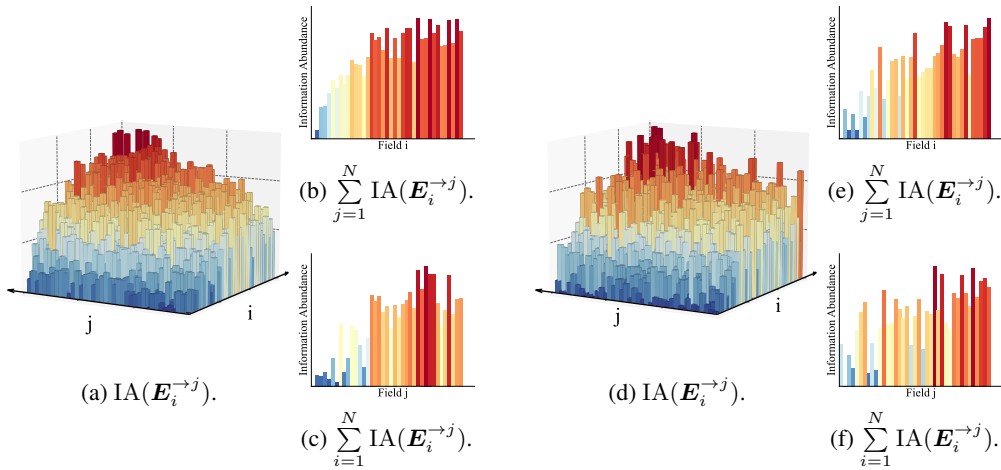

Figure 3: Visualization of information abundance of sub-embedding matrices for FFM (left) and DCNv2 (right), with field indices sorted by information abundance of corresponding raw embedding matrices. Higher or warmer indicates larger. It is observed that $\text{IA}(\boldsymbol{E}_i^{\to j})$ are co-influenced by both $\text{IA}(\boldsymbol{E}_i)$ and $\text{IA}(\boldsymbol{E}_j)$.

of information abundance, i.e., $i$ is ordered according to $\text{IA}(\boldsymbol{E}_i)$, similar to $j$. We can observe that $\text{IA}(\boldsymbol{E}_i^{\to j})$ is approximately increasing along $i$, which is trivial since $\boldsymbol{E}_i^{\to j}$ is simply a split of $\boldsymbol{E}_i$. Interestingly, another correlation can be observed that the information abundance of sub-embeddings is co-influenced by the fields it interacts with, reflected by the increasing trend along $j$, especially with larger $i$. This is amazing in the sense that even using independent embeddings to represent the same field features, these embeddings get different information abundance after learning. For instance, we calculate the summation of $\text{IA}(\boldsymbol{E}_i^{\to j})$ over $j$ or $i$ to study the effect of the other single variable, shown in Figure 3b and Figure 3c. Both of them show an increasing trend, confirming the co-influence of $i$ and $j$.

**Evidence II: Experiments on DCNv2.** An improved deep & cross network (DCNv2) (Wang et al., 2021) incorporates a crossing network which is parameterized with transformation matrices $\boldsymbol{W}_{i\to j}$ (Sun et al., 2021) over each field pair to project an embedding vector from field $i$ before interaction with field $j$. By collecting all projected embedding vectors, DCNv2 can be regarded to implicitly generate field-aware sub-embeddings $\boldsymbol{E}_i^{\to 1}, \boldsymbol{E}_i^{\to 2}, ..., \boldsymbol{E}_i^{\to N}$ to interact with all fields from embedding matrix $\boldsymbol{E}_i$, with

$$\boldsymbol{E}_i^{\to j} = \boldsymbol{E}_i \boldsymbol{W}_{i\to j}^{\top}.$$

DCNv2 consists of multiple stacked cross layers, and for simplification, we only discuss the first layer throughout this paper. Similar to Evidence I, we calculate $\text{IA}(\boldsymbol{E}_i^{\to j})$ together with the axis-wise summations and show them in the right part of Figure 3. Consistent with previous observation as FFM, the information abundance of sub-embedding matrices approximately increases along $j$ with the same $i$, even though they are projected from the same embedding matrix $\boldsymbol{E}_i$.

FIX

**Theoretical analysis: Collapse on non-sub-embedding-based models.** We now present how collapse is caused by feature interaction in non-sub-embedding-based recommendation models from a theoretical view. For simplicity, we consider an FM-style (Rendle, 2010) feature interaction. Formally, the interaction process is defined by

$$h = \sum_{i=1}^{N}\sum_{j=1}^{i-1} \boldsymbol{e}_i^{\top} \boldsymbol{e}_j = \sum_{i=1}^{N}\sum_{j=1}^{i-1} \mathbf{1}_{x_i}^{\top} \boldsymbol{E}_i \boldsymbol{E}_j^{\top} \mathbf{1}_{x_j},$$

where $h$ is the combined feature as mentioned before. Without loss of generality, we discuss one specific row $\boldsymbol{e}_1$ of $\boldsymbol{E}_1$ and keep other embedding matrices fixed. Consider a minibatch with batch size $B$. Denote $\sigma_{i,k}$ as the $k$-th singular value of $\boldsymbol{E}_i$, similar for $\boldsymbol{u}_{i,k}, \boldsymbol{v}_{i,k}$. We have

$$\begin{aligned}
\frac{\partial \mathcal{L}}{\partial \boldsymbol{e}_1} &= \frac{1}{B}\sum_{b=1}^{B}\frac{\partial \ell^{(b)}}{\partial h^{(b)}} \cdot \frac{\partial h^{(b)}}{\partial \boldsymbol{e}_1} = \frac{1}{B}\sum_{b=1}^{B}\frac{\partial \ell^{(b)}}{\partial h^{(b)}} \cdot \sum_{i=2}^{N}\boldsymbol{E}_i^{\top}\mathbf{1}_{x_i^{(b)}} \\
&= \frac{1}{B}\sum_{b=1}^{B}\frac{\partial \ell^{(b)}}{\partial h^{(b)}} \cdot \sum_{i=2}^{N}\sum_{k=1}^{K}\sigma_{i,k}\boldsymbol{v}_{i,k}\boldsymbol{u}_{i,k}^{\top}\mathbf{1}_{x_i^{(b)}} \\
&= \sum_{i=2}^{N}\sum_{k=1}^{K}\left(\frac{1}{B}\sum_{b=1}^{B}\frac{\partial \ell^{(b)}}{\partial h^{(b)}}\boldsymbol{u}_{i,k}^{\top}\mathbf{1}_{x_i^{(b)}}\right)\sigma_{i,k}\boldsymbol{v}_{i,k} \\
&= \sum_{i=2}^{N}\sum_{k=1}^{K}\alpha_{i,k}\sigma_{i,k}\boldsymbol{v}_{i,k} = \sum_{i=2}^{N}\boldsymbol{\theta}_i
\end{aligned}$$

The equation means that the gradient can be decomposed into field-specific terms. We analyze the component $\boldsymbol{\theta}_i$ for a certain field $i$, which is further decomposed into spectral for the corresponding embedding matrix $\boldsymbol{E}_i$.

From the form of $\boldsymbol{\theta}_i$, it is observed that $\{\alpha_{i,k}\}$ are $\boldsymbol{\sigma}_i$-agnostic scalars determined by the training data and objective function. Thus, the variety of $\boldsymbol{\sigma}_i$ significantly influences the composition of $\boldsymbol{\theta}_i$. For those larger $\sigma_{i,k}$, the gradient component $\boldsymbol{\theta}_i$ will be weighted more heavily along the corresponding spectral $\boldsymbol{v}_{i,k}$. When $\boldsymbol{E}_i$ is low-information-abundance, the components of $\boldsymbol{\theta}_i$ weigh imbalancely, resulting in the degeneration of $\boldsymbol{e}_1$. Since different $\boldsymbol{e}_1$ affects only $\alpha_{i,k}$ instead of $\sigma_{i,k}$ and $\boldsymbol{v}_{i,k}$, all rows of $\boldsymbol{E}_1$ degenerates in similar manners and finally form a collapsed matrix.

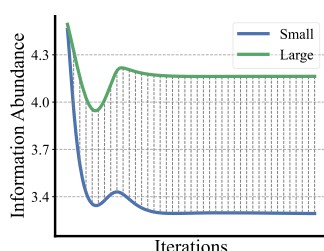

Figure 4: $\text{IA}(\boldsymbol{E}_1)$ w.r.t. training iterations for toy experiments. "Small" and "Large" refers to the cardinality of $\mathcal{X}_3$.

To further illustrate, we conduct a toy experiment over synthetic data. Suppose there are $N = 3$ fields, and we set $D_3$ to different values with $D_3 < K$ and $D_3 \gg K$ to simulate low-information-abundance and high-information-abundance cases, which matches the diverse range of the field cardinality in real-world scenarios. We train $\boldsymbol{E}_1$ while keeping $\boldsymbol{E}_2, \boldsymbol{E}_3$ fixed. Details of experiment setups are discussed in Appendix A. We show the information abundance of $\boldsymbol{E}_1$ along the training process for the two cases in Figure 4. It is observed that interacting with a low-information-abundance matrix will result in a collapsed embedding matrix.

**Summary: How is collapse caused in recommendation models?** Evidence I&II highlight that interacting with a field with a low-information-abundance embedding matrix will result in a more collapsed sub-embedding. By further considering the fact that sub-embeddings reflect the effect when fields interact since it originates from raw embeddings, we recognize the inherent mechanism of feature interaction to cause collapse, which is further confirmed by our theoretical analysis. We   NEW conclude the *interaction-collapse theory*:

> *Finding 1 (Interaction-Collapse Theory). In feature interaction of recommendation models, fields with low-information-abundance embeddings constrain the information abundance of other fields, resulting in collapsed embedding matrices.*

The interaction-collapse theory generally suggests that feature interaction is the primary catalyst for collapse, thereby imposing constraints on the ideal scalability.

## 4.2 IS IT SUFFICIENT TO AVOID COLLAPSE FOR SCALABILITY?

Following our discussion above, we have shown that the feature interaction process of recommendation models leads to collapse and thus limits the model scalability. We now discuss its negative proposition, i.e., whether suppressing the feature interaction to mitigate collapse leads to model scalability. To answer this question, we design the following two experiments to compare standard models and models with feature interaction suppressed.

**Evidence III: Regularization on DCNv2 to mitigate collapse.** Evidence II shows that a projection $\boldsymbol{W}_{i \to j}$ is learned to adjust information abundance for sub-embeddings and thus lead to collapse[1]. We now inverstigate how surpressing such effect would result in scalability by introducing the following regularization with learnable parameter $\lambda_{ij}$      FIX

$$\ell_{reg} = \sum_{i=1}^{N} \sum_{j=1}^{N} \left\| \boldsymbol{W}_{i \to j}^{\top} \boldsymbol{W}_{i \to j} - \lambda_{ij} \boldsymbol{I} \right\|_{\mathrm{F}}^{2}$$

to regularize the projection matrix to be a multiplication of an unitary matrix. This way, $\boldsymbol{W}_{i \to j}$ will preserve all normalized singular values and maintain the information abundance after projection. We experiment with various embedding sizes and compare the changes in performance, the information abundances, and the optimization dynamics for standard and regularized models. Results are shown in Figure 5. As anticipated, regularization in DCNv2 helps learn embeddings with higher information abundance. Nevertheless, from the performance perspective, the model presents unexpected results whereby the scalability does not improve or worsen as the collapse is alleviated. We further find that such a model overfits during the learning process, with the training loss consistently decreasing and the validation AUC dropping.

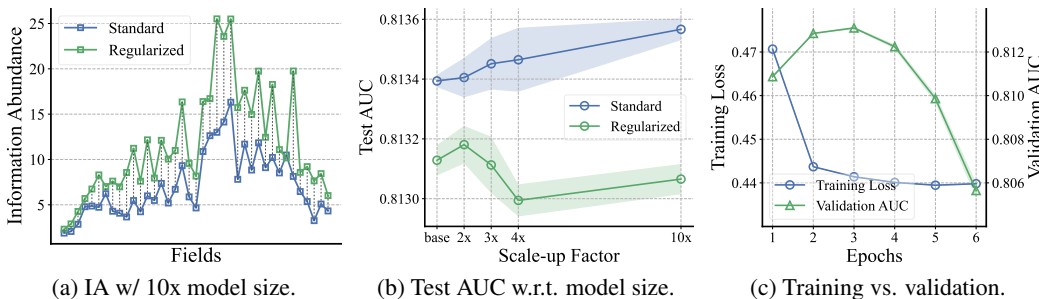

(a) IA w/ 10x model size.      (b) Test AUC w.r.t. model size.      (c) Training vs. validation.

Figure 5: Experimental results of Evidence III. Restricting DCNv2 leads to higher information abundance, yet the model suffers from over-fitting, thus resulting in non-scalability.

**Evidence IV: Scaling up DCNv2 and DNN.** We now discuss DNN, which consists of a plain interaction module by concatenating all feature vectors from different fields and processing with an MLP, formulized by

$$\boldsymbol{h} = G([\boldsymbol{e}_1, \boldsymbol{e}_2, ..., \boldsymbol{e}_N]).$$

Since DNN does not conduct explicit 2-order feature interaction (Rendle et al., 2020), following our previous interaction-collapse theory, it would suffer less from collapse. We compare the learned embeddings of DCNv2 and DNN and their perfor-

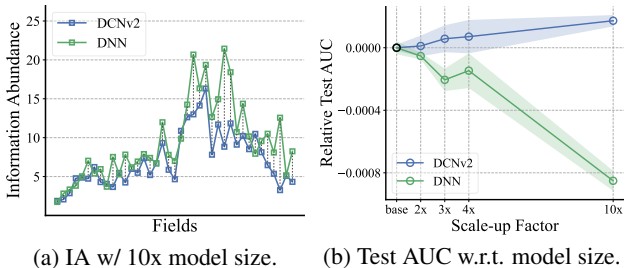

(a) IA w/ 10x model size.      (b) Test AUC w.r.t. model size.

Figure 6: Experimental results of Evidence IV. Despite higher information abundance, the performance of DNN drops w.r.t. model size.

mance with the growth of embedding size. Considering that different architectures or objectives may differ in modeling, we mainly discuss the performance trend as a fair comparison. Results are shown in Figure 6. DNN learns less-collapsed embedding matrices, reflected by higher information abundance than DCNv2. Yet, perversely, the AUC of DNN drops when increasing the embedding size, while DCNv2 sustains the performance. Such observations show that DNN falls into the issue of overfitting and lacks scalability, even though it suffers less from collapse.

**Summary: Does suppressing collapse definitely improve scalability?** Regularized DCNv2 and DNN are both models with feature interaction suppressed, and they learn less-collapsed embedding matrices than DCNv2, as expected. Yet observations in evidence III&IV demonstrate that regularized DCNv2 and DNN are both non-scalable with the growth of model size and suffer from serious overfitting. We conclude the following finding:

---

[1]Further explanation is referred to Appendix F

> *Finding 2. A less-collapsed model with feature interaction suppressed is insufficient for scalability due to overfitting concern.*

Such a finding is plausible, considering that feature interaction brings domain knowledge of higher-order correlations in recommender systems and helps form generalizable representations. When feature interaction is suppressed, models tend to fit noise as the embedding size increases, resulting in reduced generalization.

## 5 MULTI-EMBEDDING DESIGN

In this section, we present a simple design of *multi-embedding*, which serves as an effective scaling design applicable to a wide range of model architecture designs. We introduce the overall architecture, present experimental results, and analyze how multi-embedding works.

### 5.1 MULTI-EMBEDDING FOR BETTER SCALABILITY

The two-sided effect of feature interaction for scalability implies a *principle* for model design. That is, a scalable model should be capable of less-collapsed embeddings within the existing feature interaction framework instead of removing interaction. Based on this principle, we propose *multi-embedding* or *ME* as a simple yet efficient design to improve scalability. Specifically, we scale up the number of independent and complete embedding sets instead of the embedding size, and incorporate embedding-set-specific feature interaction layers. Similar to previous works such as group convolution (Krizhevsky et al., 2012), multi-head attention (Vaswani et al., 2017), and other ~~NEW~~ decoupling-based works in recommender systems (Liu et al., 2022; 2019; Weston et al., 2013), such design allows the model to learn different interaction patterns jointly, while a single-embedding model would be limited to the only interaction pattern that causes severe collapse. This way, the model is capable of learning diverse embedding vectors to mitigate collapse while keeping the original interaction modules. Formally, a model with $M$ sets of embeddings is defined as

$$\boldsymbol{e}_i^{(m)} = \left(\boldsymbol{E}_i^{(m)}\right)^\top \mathbf{1}_{x_i}, \ \forall i \in \{1, 2, ..., N\},$$

$$h^{(m)} = I^{(m)} \left(\boldsymbol{e}_1^{(m)}, \boldsymbol{e}_2^{(m)}, ..., \boldsymbol{e}_N^{(m)}\right),$$

$$h = \frac{1}{M} \sum_{m=1}^M h^{(m)}, \quad \hat{y} = F(h),$$

where $m$ stands for the index of embedding set. One requirement of multi-embedding is that there should be non-linearities such as ReLU in interaction $I$; otherwise, the model is equivalent to single-embedding and hence does not capture different patterns (see Appendix B). As a solution, we add a non-linear projection after interaction for the model with linear interaction layers and reduce one MLP layer for $F$ to achieve a fair comparison. An overall architecture comparison of single-embedding and mult-embedding models with $N = 2$ and $M = 2$ is shown in Figure 7.

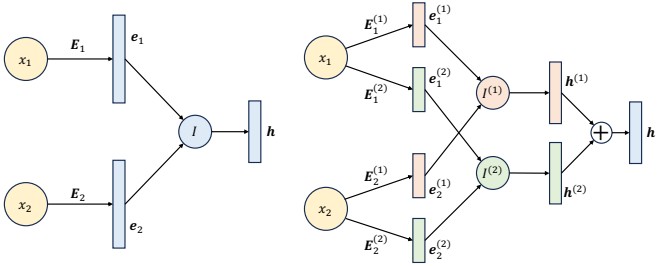

Figure 7: Architectures of single-embedding (left) and multi-embedding (right) models with $N = 2$ and $M = 2$.

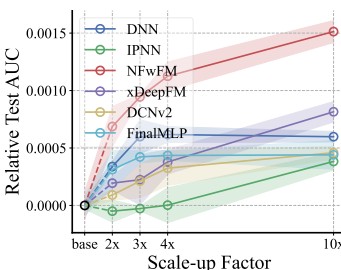

Figure 8: Scalability of multi-embedding on Criteo dataset.

## 5.2 EXPERIMENTS

**Setup.** We conduct our experiments on two datasets for recommender systems: Criteo (Jean-Baptiste Tien, 2014) and Avazu (Steve Wang, 2014), which are large and hard benchmark datasets widely used in recommender systems. We experiment on baseline models including DNN, IPNN (Qu et al., 2016), NFwFM (Pan et al., 2018), xDeepFM (Lian et al., 2018), DCNv2 (Wang et al., 2021), FinalMLP (Mao et al., 2023) and their corresponding multi-embedding variants with 2x, 3x, 4x and 10x model size[2]. Here NFwFM is a variant of NFM He & Chua (2017) by replacing FM with FwFM. All experiments are performed with 8/1/1 training/validation/test splits, and we apply early stopping based on validation AUC. More details are shown in Appendix C.2.

**Results.** We repeat each experiment 3 times and report the average test AUC with different scaling factors of the model size. Results are shown in Table 1. For the experiments with single-embedding, we observe that all the models demonstrate poor scalability. Only DCNv2 and NFwFM show slight improvements with increasing embedding sizes, with gains of 0.00036 on Criteo and 0.00090 on Avazu, respectively. For DNN, xDeepFM, and FinalMLP, which rely highly on non-explicit interaction, the performance even drops (0.00136 on Criteo and 0.00118 on Avazu) when scaled up to 10x, as discussed in Section 4.2. In contrast to single-embedding, our multi-embedding shows consistent and remarkable improvement with the growth of the embedding size, and the highest performance is always achieved with the largest 10x size. For DCNv2 and NFwFM, multi-embedding gains 0.00099 on Critio and 0.00202 on Avazu by scaling up to 10x, which is never obtained by single-embedding. Over all models and datasets, compared with baselines, the largest models averagely achieve 0.00106 improvement on the test AUC[3]. Multi-embedding provides a methodology to break through the non-scalability limit of existing models. We visualize the scalability of multi-embedding on Criteo dataset in Figure 8. The standard deviation and detailed scalability comparison are shown in Appendix C.3.

Table 1: Test AUC for different models. Higher indicates better. Underlined and bolded values refer to the best performance with single-embedding (SE) and multi-embedding (ME), respectively.

| Model | | Criteo | | | | | Avazu | | | | |
|---|---|---|---|---|---|---|---|---|---|---|---|
| | | base | 2x | 3x | 4x | 10x | base | 2x | 3x | 4x | 10x |
| DNN | SE | 0.81228 | 0.81222 | 0.81207 | 0.81213 | 0.81142 | 0.78744 | 0.78759 | 0.78752 | 0.78728 | 0.78648 |
| | ME | | 0.81261 | **0.81288** | **0.81289** | **0.81287** | | 0.78805 | 0.78826 | 0.78862 | **0.78884** |
| IPNN | SE | 0.81272 | 0.81273 | 0.81272 | 0.81271 | 0.81262 | 0.78732 | 0.78741 | 0.78738 | 0.78750 | 0.78745 |
| | ME | | 0.81268 | 0.81270 | 0.81273 | **0.81311** | | 0.78806 | 0.78868 | 0.78902 | **0.78949** |
| NFwFM | SE | 0.81059 | 0.81087 | 0.81090 | 0.81112 | 0.81113 | 0.78684 | 0.78757 | 0.78783 | 0.78794 | – |
| | ME | | 0.81128 | 0.81153 | 0.81171 | **0.81210** | | 0.78868 | 0.78901 | **0.78932** | – |
| xDeepFM | SE | 0.81217 | 0.81180 | 0.81167 | 0.81137 | 0.81116 | 0.78743 | 0.78750 | 0.78714 | 0.78735 | 0.78693 |
| | ME | | 0.81236 | 0.81239 | 0.81255 | **0.81299** | | 0.78848 | 0.78886 | 0.78894 | **0.78927** |
| DCNv2 | SE | 0.81339 | 0.81341 | 0.81345 | 0.81346 | 0.81357 | 0.78786 | 0.78835 | 0.78854 | 0.78852 | 0.78856 |
| | ME | | 0.81348 | 0.81361 | **0.81382** | **0.81385** | | 0.78862 | 0.78882 | 0.78907 | **0.78942** |
| FinalMLP | SE | 0.81259 | 0.81262 | 0.81248 | 0.81240 | 0.81175 | 0.78751 | 0.78797 | 0.78795 | 0.78742 | 0.78662 |
| | ME | | 0.81290 | **0.81302** | **0.81303** | **0.81303** | | 0.78821 | **0.78831** | **0.78836** | **0.78830** |

## 5.3 ANALYSIS

**Information abundance.** Multi-embedding models achieve remarkable scalability compared with single-embedding. We verify that such scalability originates from the mitigation of collapse. We compare the information abundance of single-embedding and multi-embedding DCNv2 with the 10x embedding size. As shown in Figure 9a, multi-embedding offers higher information abundance and indicates less collapsed embedding matrices.

**Variations of embeddings.** Multi-embedding utilizes embedding-set-specific interactions to enrich embedding learning. We analyze the information abundance for each embedding set as shown

---

[2]The embedding of NFwFM with 10x size on Avazu costs nearly 37.6GB memory, which exceeds our GPU memory limit. Therefore, we do not conduct 10x NFwFM on Avazu. On the other hand, the existing experiment with 4x is already sufficient for NFwFM on Avazu.

[3]A slightly higher AUC at 0.001-level is regarded significant (Cheng et al., 2016; Guo et al., 2017; Song et al., 2019; Tian et al., 2023)

in Figure 9b. It is observed that the embedding matrices of different sets vary in information abundance.

**Different interaction patterns.** To justify that the scalability of multi-embedding originates from different interaction patterns, we visualize $\|\boldsymbol{W}_{i\to j}^{(m)}\|_{\mathrm{F}}$ as the interaction pattern (Wang et al., 2021) for a multi-embedding DCNv2 model in Figure 9c. It is shown that the interaction layers learn various patterns. To further illustrate, we conduct an ablation study by restricting the divergence of $\|\boldsymbol{W}_{i\to j}^{(m)}\|_{\mathrm{F}}$ across all embedding sets. From results in Figure 9d, it is observed that the divergence-restricted multi-embedding model does not show similar scalability as standard multi-embedding models, indicating multi-embedding works from the diversity of interaction layers. Ablation study    NEW
on sharing one interaction layer across all embedding sets are provided in Appendix H.

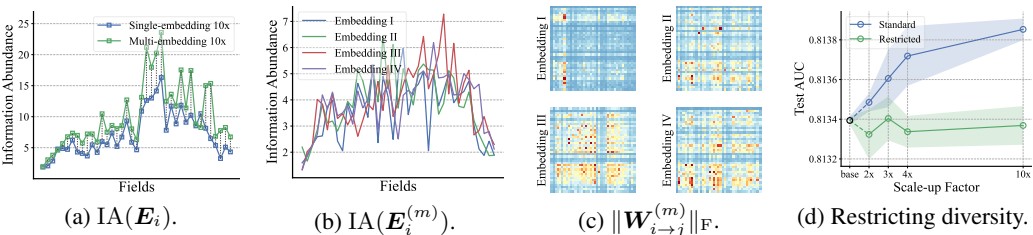

(a) IA($\boldsymbol{E}_i$).  (b) IA($\boldsymbol{E}_i^{(m)}$).  (c) $\|\boldsymbol{W}_{i\to j}^{(m)}\|_{\mathrm{F}}$.  (d) Restricting diversity.

Figure 9: Analysis of multi-embedding. **(a)**: Multi-embedding learns higher information abundance. **(b)**: Each embedding set learns diverse embeddings, relected by varying information abundance. **(c)**: Embedding-set-specific feature interaction layers capture different interaction patterns. **(d)**: Restricting diversity of $\|\boldsymbol{W}_{i\to j}^{(m)}\|_{\mathrm{F}}$ across all embedding sets leads to non-scalability.

## 6 RELATED WORKS

**Modules in recommender systems.** Plenty of existing works investigate the module design for recommender systems. A line of studies focuses on feature interaction process (Koren et al., 2009; Rendle, 2010; Juan et al., 2016; Qu et al., 2016; He & Chua, 2017; Guo et al., 2017; Pan et al., 2018; Lian et al., 2018; Song et al., 2019; Cheng et al., 2020; Sun et al., 2021; Wang et al., 2021; Mao et al., 2023; Tian et al., 2023), which is specific for recommender systems. These works are built up to fuse domain-specific knowledge of recommender systems. In contrast to proposing new modules, our work starts from a view of machine learning and analyzes the existing models for scalability.

**Collapse phenomenon.** Neural collapse or representation collapse describes the degeneration of representation vectors with restricted variation. This phenomenon is widely studied in supervised learning (Papyan et al., 2020; Zhu et al., 2021; Tirer & Bruna, 2022), unsupervised contrastive learning (Hua et al., 2021; Jing et al., 2021; Gupta et al., 2022), transfer learning (Aghajanyan et al., 2020; Kumar et al., 2022) and generative models (Mao et al., 2017; Miyato et al., 2018). Chi et al. (2022) discuss the representation collapse in sparse MoEs. Inspired by these works, we realize the embedding collapse of recommendation models when regarding embedding vectors as representations by their definition, yet we are facing the setting of field-level interaction, which has not previously been well studied.

**Intrinsic dimensions and compression theories.** To describe the complexity of data, existing works include intrinsic-dimension-based quantification (Levina & Bickel, 2004; Ansuini et al., 2019; Pope et al., 2020) and pruning-based analysis (Wen et al., 2017; Alvarez & Salzmann, 2017; Sun et al., 2021). Our SVD-based concept of information abundance is related to these works.

## 7 CONCLUSION

In this paper, we highlight the non-scalability issue of existing recommendation models and identify the embedding collapse phenomenon that hinders scalability. From empirical and theoretical analysis around embedding collapse, we conclude the two-sided effect of feature interaction on scalability, i.e., feature interaction causes collapse while reducing overfitting. We propose a unified design of multi-embedding to mitigate collapse without suppressing feature interaction. Experiments on benchmark datasets demonstrate that multi-embedding consistently improves model scalability.

REPRODUCBILITY STATEMENT

For toy experiments, we show the detailed settings in Appendix A. For experiments on benchmark datasets, we follow the default data pre-processing according to the repository of pytorch-fm[4]. We present the general model architecture in Section 5.1, and demonstrate the specific design and all hyperparameters in Appendix C.2. We show the confidence of results with empirical standard deviations in Appendix C.3. We will release our code in case our paper is accepted.

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

## A   DETAILS OF TOY EXPERMIENT

In this section, we present the detailed settings of the toy experiment. We consider a scenario with $N = 3$ fields and $D_1 = D_2 = 100$. For each $(x_1, x_2) \in \mathcal{X}_1 \times \mathcal{X}_2$, we randomly assign $x_3 \sim \mathcal{U}[\mathcal{X}_3]$, $y \sim \mathcal{U}\{0, 1\}$ and let $(\boldsymbol{x}, y)$ to be one piece of data, thus for different values of $D_3$, there are always $100^2$ pieces of data, and they follow the same distribution when reduced on $\mathcal{X}_1 \times \mathcal{X}_2$. We set $D_3 = 3$ and $D_3 = 100$ to simulate the case with low-information-abundance and high-information-abundance, respectively. We randomly initialize all embedding matrices with normal distribution $\mathcal{N}(0, 1)$, fix $\boldsymbol{E}_2, \boldsymbol{E}_3$ and only optimize $\boldsymbol{E}_1$ during training. We use full-batch SGD with the learning rate of 1. We train the model for 5,000 iterations in total.

## B   NON-LINEARITY FOR MULTI-EMBEDDING

We have mentioned that the embedding-set-specific feature interaction of multi-embedding should contain non-linearity, otherwise the model will degrade to a single-embedding model. For simplicity, we consider a stronger version of multi-embedding, where the combined features from different embedding sets are concatenated instead of averaged. To further illustrate, consider linear feature interaction modules $I^{(m)} : \left(\mathbb{R}^K\right)^N \to \mathbb{R}^h$, then we can define a linear feature interaction module $I_{\text{all}} : \left(\mathbb{R}^{MK}\right)^N \to \mathbb{R}^{Mh}$. For convenience, we denote $[f(i)]_{i=1}^n$ as $[f(1), f(2), ..., f(n)]$, and $\boldsymbol{e}_i = [e_i^m]_{m=1}^M$. The form of $I_{\text{all}}$ can be formulated by

$$I_{\text{all}}\left(\boldsymbol{e}_1, \boldsymbol{e}_2, ..., \boldsymbol{e}_N\right) = \left[I^{(m)}(\boldsymbol{e}_1^{(m)}, ..., \boldsymbol{e}_N^{(m)})\right]_{m=1}^M.$$

This shows a multi-embedding model is equivalent to a model by concatenating all embedding sets. We will further show that the deduced model with $I_{\text{all}}$ is homogeneous to a single-embedding model with size $MK$, i.e., multi-embedding is similar to single-embedding with linear feature interaction modules. Denote the feature interaction module of single-embedding as $I$. Despite $I_{\text{all}}$ could have different forms from $I$, we further give three examples to show the homogeneity of $I_{\text{all}}$ and $I$.

**DNN.**   Ignoring the followed MLP, DNN incorporate a non-parametric interaction module by concatenating all fields together. Formally, we have

$$I(\boldsymbol{e}_1, ..., \boldsymbol{e}_N) = \left[[e_i^{(m)}]_{m=1}^M\right]_{i=1}^N,$$

$$I_{\text{all}}(\boldsymbol{e}_1, ..., \boldsymbol{e}_N) = \left[[e_i^{(m)}]_{i=1}^N\right]_{m=1}^M.$$

In other words, $I$ and $I_{\text{all}}$ only differ in a permutation, thus multi-embedding and single-embedding are equivalent.

**Projected DNN.**   If we add a linear projection after DNN, then we can split the projection for fields and embedding sets, and derive

$$I(\boldsymbol{e}_1, ..., \boldsymbol{e}_N) = \sum_{i=1}^N \sum_{m=1}^M \boldsymbol{W}_{i,m} e_i^{(m)},$$

$$I_{\text{all}}(\boldsymbol{e}_1, ..., \boldsymbol{e}_N) = \left[\sum_{i=1}^N \boldsymbol{W}_{i,m} e_i^{(m)}\right]_{m=1}^M.$$

In other words, $I$ and $I_{\text{all}}$ only differ in a summation. Actually if we average the combined features for $I_{\text{all}}$ rather than concatenate to restore our proposed version of multi-embedding, then multi-embedding and single-embedding are equivalent by the scalar $1/M$.

**DCNv2.**   DCNv2 incorporates the following feature interaction by

$$I(\boldsymbol{e}_1, ..., \boldsymbol{e}_N) = \left[\boldsymbol{e}_i \odot \sum_{j=1}^N \boldsymbol{W}_{j \to i} \boldsymbol{e}_j\right]_{i=1}^N,$$

thus by splitting $\boldsymbol{W}_{i \to j}$ we have

$$I(\boldsymbol{e}_1, ..., \boldsymbol{e}_N) = \left[ [\boldsymbol{e}_i^{(m)} \odot \sum_{j=1}^{N} \sum_{m'=1}^{M} \boldsymbol{W}_{j \to i}^{(m,m')} \boldsymbol{e}_j^{(m')}]_{m=1}^{M} \right]_{i=1}^{N}$$

$$I_{\text{all}}(\boldsymbol{e}_1, ..., \boldsymbol{e}_N) = \left[ [\boldsymbol{e}_i^{(m)} \odot \sum_{j=1}^{N} \boldsymbol{W}_{j \to i}^{(m)} \boldsymbol{e}_j^{(m)}]_{i=1}^{N} \right]_{m=1}^{M}.$$

By simply letting $\boldsymbol{W}^{(m,m)} = \boldsymbol{W}^{(m)}$ and $\boldsymbol{W}^{(m,m')} = \boldsymbol{O}$ for $m \neq m'$, we convert a multi-embedding model into a single-embedding model under permutation. Therefore, multi-embedding is a special case of single-embedding for DCNv2.

**Summary.** In summary, a linear feature interaction module will cause homogeneity between single-embedding and multi-embedding. Hence it is necessary to use or introduce non-linearity in feature interaction module.

## C  DETAILS OF EXPERIMENT

### C.1  DATASET DESCRIPTION

The statistics of Criteo and Avazu are shown in Table 2. It is shown that the data amount is ample and $D_i$ can vary in a large range.

Table 2: Statistics of benchmark datasets for experiments.

| Dataset | #Instances | #Fields | $\sum_i D_i$ | $\max\{D_i\}$ | $\min\{D_i\}$ |
|---------|-----------|---------|--------------|---------------|---------------|
| Criteo  | 45.8M     | 39      | 1.08M        | 0.19M         | 4             |
| Avazu   | 40.4M     | 22      | 2.02M        | 1.61M         | 5             |

### C.2  EXPERIMENT SETTINGS

**Specific multi-embedding design.** For DCNv2, DNN, IPNN and NFwFM, we add one non-linear projection after the stacked cross layers, the concatenation layer, the inner product layer and the field-weighted dot product layer, respectively. For xDeepFM, we directly average the output of the compressed interaction network, and process the ensembled DNN the same as the pure DNN model. For FinalMLP, we average the two-stream outputs respectively.

**Hyperparameters.** For all experiments, we split the dataset into $8 : 1 : 1$ for training/validation/test with random seed 0. We use the Adam optimizer with batch size 2048, learning rate 0.001 and weight decay 1e-6. For base size, we use embedding size 50 for NFwFM considering the pooling, and 10 for all other experiments. We find the hidden size and depth of MLP does not matters the result, and for simplicity, we set hidden size to 400 and set depth to 3 (2 hidden layers and 1 output layer) for all models. We use 4 cross layers for DCNv2 and hidden size 16 for xDeepFM. All experiments use early stopping on validation AUC with patience 3. We repeat each experiment for 3 times with different random initialization. All experiments can be done with a single NVIDIA GeForce RTX 3090.

### C.3  EXPERIMENTAL RESULTS

Here we present detailed experimental results with estimated standard deviation. Specifically, we show results on Criteo dataset in Table 3 and Figure 10 and Avazu dataset in Table 4 and Figure 11.

Table 3: Results on Criteo dataset. Higher indicates better.

| Model | | base | 2x | 3x | 4x | 10x |
|-------|----|------|-----|-----|-----|------|
| | | | | Criteo | | |
| DNN | SE | $0.81228_{\pm 0.00004}$ | $0.81222_{\pm 0.00002}$ | $0.81207_{\pm 0.00007}$ | $0.81213_{\pm 0.00011}$ | $0.81142_{\pm 0.00006}$ |
| | ME | | $0.81261_{\pm 0.00004}$ | $0.81288_{\pm 0.00015}$ | $0.81289_{\pm 0.00007}$ | $0.81287_{\pm 0.00005}$ |
| IPNN | SE | $0.81272_{\pm 0.00003}$ | $0.81273_{\pm 0.00013}$ | $0.81272_{\pm 0.00004}$ | $0.81271_{\pm 0.00007}$ | $0.81262_{\pm 0.00016}$ |
| | ME | | $0.81268_{\pm 0.00009}$ | $0.81270_{\pm 0.00002}$ | $0.81273_{\pm 0.00015}$ | $0.81311_{\pm 0.00008}$ |
| NFwFM | SE | $0.81059_{\pm 0.00012}$ | $0.81087_{\pm 0.00008}$ | $0.81090_{\pm 0.00012}$ | $0.81112_{\pm 0.00011}$ | $0.81113_{\pm 0.00022}$ |
| | ME | | $0.81128_{\pm 0.00017}$ | $0.81153_{\pm 0.00002}$ | $0.81171_{\pm 0.00012}$ | $0.81210_{\pm 0.00010}$ |
| xDeepFM | SE | $0.81217_{\pm 0.00003}$ | $0.81180_{\pm 0.00002}$ | $0.81167_{\pm 0.00008}$ | $0.81137_{\pm 0.00005}$ | $0.81116_{\pm 0.00009}$ |
| | ME | | $0.81236_{\pm 0.00006}$ | $0.81239_{\pm 0.00022}$ | $0.81255_{\pm 0.00011}$ | $0.81299_{\pm 0.00009}$ |
| DCNv2 | SE | $0.81339_{\pm 0.00002}$ | $0.81341_{\pm 0.00007}$ | $0.81345_{\pm 0.00009}$ | $0.81346_{\pm 0.00011}$ | $0.81357_{\pm 0.00004}$ |
| | ME | | $0.81348_{\pm 0.00005}$ | $0.81361_{\pm 0.00014}$ | $0.81382_{\pm 0.00015}$ | $0.81385_{\pm 0.00005}$ |
| FinalMLP | SE | $0.81259_{\pm 0.00009}$ | $0.81262_{\pm 0.00007}$ | $0.81248_{\pm 0.00008}$ | $0.81240_{\pm 0.00002}$ | $0.81175_{\pm 0.00020}$ |
| | ME | | $0.81290_{\pm 0.00017}$ | $0.81302_{\pm 0.00005}$ | $0.81303_{\pm 0.00004}$ | $0.81303_{\pm 0.00012}$ |

Figure 10: Visualization of scalability on Criteo dataset.

# D   SCALING UP OTHER MODULES OF RECOMMENDATION MODELS

NEW

For recommendation models, the embedding module occupies the largest number of parameters ($> 92\%$ in our DCNv2 baseline for Criteo, and even larger for industrial models), and thus serves as the important and informative bottleneck part of the model. To further illustrate, we discuss the scaling up of the other modules of recommendation models, i.e., the feature interaction module $I$ and the postprocessing prediction module $F$. We experiment to increase #cross layers and #MLP layers in the DCNv2 baseline and show the results in Table 5. It is observed that increasing #cross layers or #MLP layers does not lead to performance improvement, hence it is reasonable and necessary to scale up the embedding size.

# E   DISCUSSION OF EXTENSION OF INFORMATION ABUNDANCE

NEW

Our proposed information abundance is a fair metric when two embedding matrices have the same embedding size. To apply the definition between different embedding sizes, some possible extensions include $\frac{\text{IA}(\boldsymbol{E})}{K}$ and $\frac{\text{IA}(\boldsymbol{E})}{\mathbb{E}[\text{IA}(\texttt{randn\_like}(\boldsymbol{E}))]}$, where $K$ stands for the embedding size and

Table 4: Results on Avazu dataset. Higher indicates better.

| Model | | Avazu | | | | |
|---|---|---|---|---|---|---|
| | | base | 2x | 3x | 4x | 10x |
| DNN | SE | $0.78744_{\pm 0.00008}$ | $0.78759_{\pm 0.00011}$ | $0.78752_{\pm 0.00031}$ | $0.78728_{\pm 0.00036}$ | $0.78648_{\pm 0.00013}$ |
| | ME | | $0.78805_{\pm 0.00017}$ | $0.78826_{\pm 0.00013}$ | $0.78862_{\pm 0.00026}$ | $0.78884_{\pm 0.00005}$ |
| IPNN | SE | $0.78732_{\pm 0.00020}$ | $0.78741_{\pm 0.00022}$ | $0.78738_{\pm 0.00010}$ | $0.78750_{\pm 0.00007}$ | $0.78745_{\pm 0.00018}$ |
| | ME | | $0.78806_{\pm 0.00012}$ | $0.78868_{\pm 0.00023}$ | $0.78902_{\pm 0.00009}$ | $0.78949_{\pm 0.00028}$ |
| NFwFM | SE | $0.78684_{\pm 0.00017}$ | $0.78757_{\pm 0.00020}$ | $0.78783_{\pm 0.00009}$ | $0.78794_{\pm 0.00022}$ | – |
| | ME | | $0.78868_{\pm 0.00038}$ | $0.78901_{\pm 0.00029}$ | $0.78932_{\pm 0.00035}$ | – |
| xDeepFM | SE | $0.78743_{\pm 0.00009}$ | $0.78750_{\pm 0.00025}$ | $0.78714_{\pm 0.00030}$ | $0.78735_{\pm 0.00004}$ | $0.78693_{\pm 0.00050}$ |
| | ME | | $0.78848_{\pm 0.00006}$ | $0.78886_{\pm 0.00026}$ | $0.78894_{\pm 0.00004}$ | $0.78927_{\pm 0.00019}$ |
| DCNv2 | SE | $0.78786_{\pm 0.00022}$ | $0.78835_{\pm 0.00023}$ | $0.78854_{\pm 0.00010}$ | $0.78852_{\pm 0.00003}$ | $0.78856_{\pm 0.00016}$ |
| | ME | | $0.78862_{\pm 0.00011}$ | $0.78882_{\pm 0.00012}$ | $0.78907_{\pm 0.00011}$ | $0.78942_{\pm 0.00024}$ |
| FinalMLP | SE | $0.78751_{\pm 0.00026}$ | $0.78797_{\pm 0.00019}$ | $0.78795_{\pm 0.00017}$ | $0.78742_{\pm 0.00015}$ | $0.78662_{\pm 0.00025}$ |
| | ME | | $0.78821_{\pm 0.00013}$ | $0.78831_{\pm 0.00029}$ | $0.78836_{\pm 0.00018}$ | $0.78830_{\pm 0.00022}$ |

Figure 11: Visualization of scalability on Avazu dataset.

`randn_like`($\boldsymbol{E}$) refers to a random matrix underlying the normal distribution with the same shape as $\boldsymbol{E}$. We compare the former one with different embedding sizes in Figure 12, and it is shown that the degree of collapse increases with respect to the embedding size, which is consistent with the observation in Figure 1b

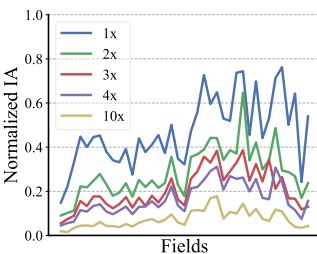

Figure 12: Normalized information abundance $\frac{\text{IA}(\boldsymbol{E})}{K}$ for different embedding sizes on DCNv2.

Table 5: Test AUC with enlarged feature interaction modules and portprocessing prediction modules. Higher indicates better.

| DCNv2 | 1x | 2x | 4x |
|---|---|---|---|
| standard | 0.81339 | 0.81341 | 0.81346 |
| + #cross layer | 0.81325 | 0.81338 | 0.81344 |
| + #MLP depth | 0.81337 | 0.81345 | 0.81342 |

## F  DETAILED EXPLANATION OF REGULARIZED DCNV2

NEW

Regarding Evidence II, we proposed regularization of the weight matrix $W_i^{\to j}$ to mitigate the collapse caused by the projection $W_i^{\to j}$ in sub-embeddings. By regularizing $W_i^{\to j}$ to be a unitary matrix (or the multiplication of unitary matrices), we ensure the preservation of all singular values of the sub-embedding. Consequently, the information abundance of sub-embeddings in regularized DCNv2 is larger than standard DCNv2. We plot the heatmap of information abundance of embeddings and sub-embeddings Figure 13. This clearly demonstrates that regularized DCNv2 exhibits a higher information abundance. Based on our Finding 1, regularized DCNv2 mitigates the problem of embedding collapse by increasing the information abundance of the sub-embeddings that directly interact with the embeddings.

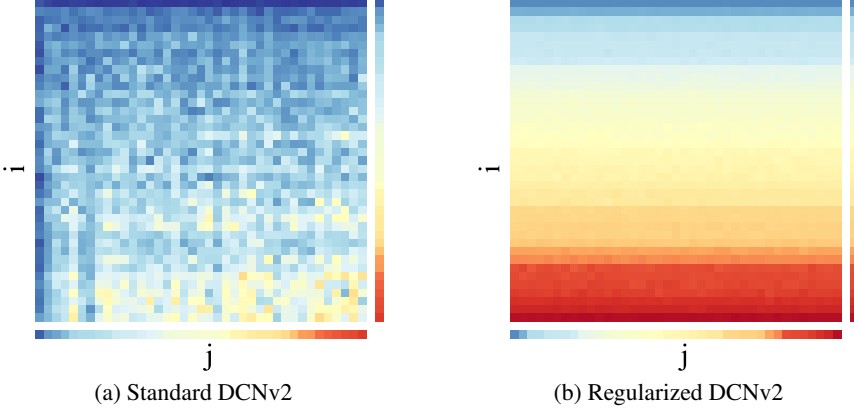

(a) Standard DCNv2        (b) Regularized DCNv2

Figure 13: Visualization of information abundance of embeddings and sub-embeddings for standard and regularized DCNv2, respectively. The rightmost and downmost bars correspond to $\mathrm{IA}(E_i)$ or $\mathrm{IA}(E_j)$. Compared with the standard DCNv2, the regularized can preserve singular values, and resulting in less-collapsed sub-embeddings and finally larger information abundance as in Figure 5a.

## G  MORE ANALYSIS OF ME

NEW

In this section we present further analysis of ME. We conduct the experiments of sub-embedding analysis of **Evidence II** and Regularized DCNv2 of **Evidence III** in Section 4.

**Evidence II for ME.**  We calculate the information abundance of all the sub-embeddings in an ME DCNv2 model, where the sub-embedding is collected by concatenating the projected embeddings from each embedding set, i.e.,

$$E_i^{\to j} = \left[ E_i^{(1)}(W_{i,j}^{(1)})^\top, ..., E_i^{(M)}(W_{i,j}^{(M)})^\top \right],$$

and compare them with that of SE in Figure 14. It can be observed that $\mathrm{IA}(E_i^{\to j})$ is less influenced by the field $j$ to interact with, especially when comparing Figure 14c and 14f, indicating that ME helps to mitigate the collapse caused by feature interaction.

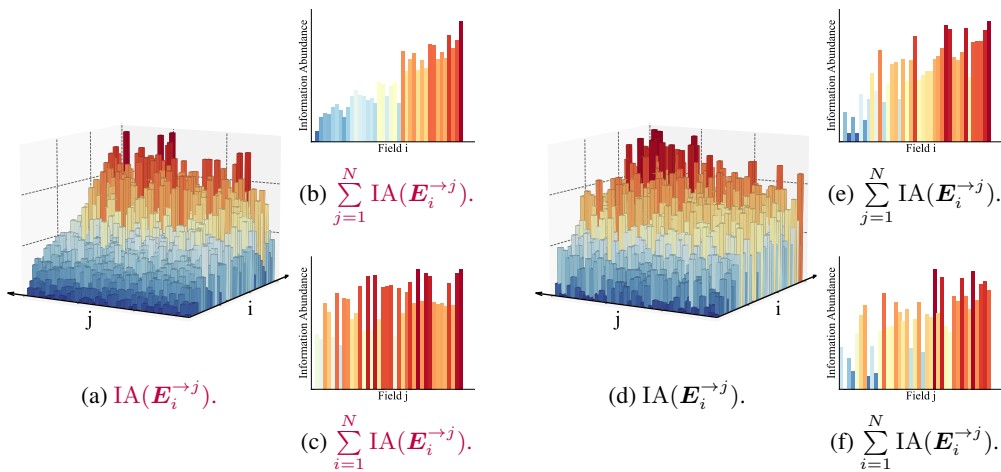

Figure 14: Visualization of information abundance of sub-embedding matrices for ME (left) and SE (right) on DCNv2, with field indices sorted by information abundance of corresponding raw embedding matrices. Higher or warmer indicates larger. It is observed that $\text{IA}(\boldsymbol{E}_i^{\to j})$ are less influenced by $\text{IA}(\boldsymbol{E}_j)$ in ME.

**Evidence III for ME.** We add regularization

$$\ell_{reg} = \sum_{m=1}^{M} \sum_{i=1}^{N} \sum_{j=1}^{N} \left\| (\boldsymbol{W}_{i \to j}^{(m)})^{\top} \boldsymbol{W}_{i \to j}^{(m)} - \lambda_{ij}^{(m)} \boldsymbol{I} \right\|_{\text{F}}^2$$

for ME DCNv2 and conduct experiments with different embedding sizes. Results are shown in Figure 15. It is observed that the regularized ME DCNv2 model exhibits a similar increasing trend in performance with respect to the embedding size as the regularized SE model. However, the performance decline in the regularized ME model is less significant compared to the regularized SE model. This finding provides further evidence that ME provides model scalability even when applied on an interaction layer to cause overfitting.

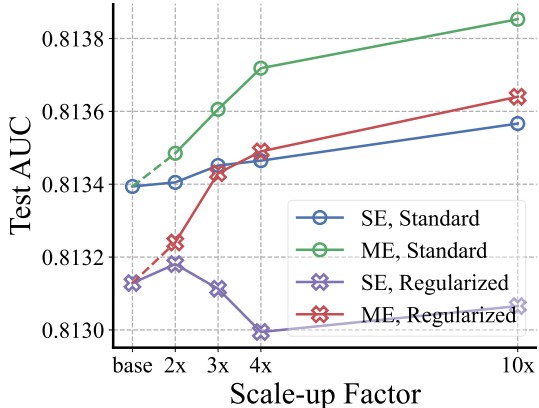

Figure 15: Test AUC w.r.t. model size.

## H ABLATION STUDY ON ME

NEW

In Figure 9d, we present an ablation study by implicitly aligning the norms of weights across all interaction modules of different embedding sets. Here we give a simpler and more straightforward ablation experiment by sharing only one interaction module for all embedding sets. Results are shown in Figure 16. It is evident that ME with shared interaction performs worse than ME with specific interactions (and even worse than the single embedding, SE). Such finding is consistent with our analysis that ME ¡u¿works from the diversity of interaction layers.

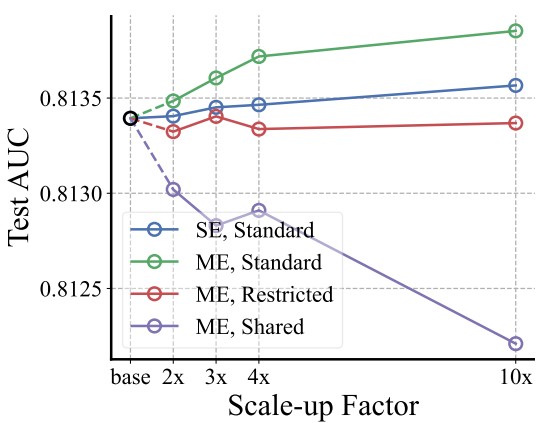

Figure 16: Test AUC w.r.t. model size.

