# OpenReview forum: "On the Embedding Collapse When Scaling up Recommendation Models"
_ICLR.cc/2024/Conference — Submitted to ICLR 2024_

### Official Review · Reviewer_eFdv · 2023-10-31

**Soundness:** 3 good
**Presentation:** 3 good
**Contribution:** 2 fair
**Rating:** 6
**Confidence:** 3

**Summary:**

This paper observes scaling the embedding dimensionality does not lead to satisfactory improvement in recommendation models and claims this is due to the dimensional collapse problem where the learned embedding vector spans only on lower dimensional subspace. The paper further claims such problem will propagate by feature interaction module when a feature embedding interacts with collapsed embeddings and regularization could mitigate collapse but harms the performance. Thus this paper proposes an alternative approach to address the embedding collapse issue, namely to replace single embedding in the original model with multiple embeddings. The paper shows the proposed approach lead to higher "information abundance" which is ratio between sum of absolute singular values and the largest singular value, indicate the "spreadness"/concentration of the singular value distribution. The paper also applied the proposed approach on several recommender models on two datasets, shows a absolute improvement of AUC on 1e-3 ~ 1e-4 level, when scaling the number of multi-embeddings.

initial recommendation: weak reject, for reasons please see the weak points.

**Strengths:**

* The claim about the dimensional collapsing problem when scaling the dimensionality of feature embeddings in recommender system is reasonably supported in the paper and sound.
* The analysis about the trade off between feature embedding collapsing and the regularization level of feature interaction is interesting and reasonable.
* The paper is easy to follow

**Weaknesses:**

* The computation of "information abundance" for multi-embedding setting is not clearly defined in the paper. As show in the Figure 1.b for features with low predictive power would have embedding with low "information abundance" ratio, and scaling the dimensionality would further lower the ratio. Thus if the ratio for multi embedding is computed by averaging over the small embeddings, the lower ratios for some small embeddings will be compensated. This possibility makes the proposed abundance ratio less reliable.
* The idea of multi-facet embedding or polysemy embedding has been studied quite extensively in the past. From network embedding (Liu et al. Is a single vector enough? exploring node polysemy for network embedding) to recommender systems (Weston et al. Nonlinear latent factorization by embedding multiple user interests). However, non of the related work on multi-embedding has been discussed in the paper.
* The finding 1 was claimed to "applicable to general recommendation models instead of only the models with sub-embeddings" without being linked to any evidences. Also, I personally find find 1 is much a theory rather than a "law".

**Questions:**

* How is the "information abundance" for multi-embeddings computed?
* How would the proposed multi-embedding approach be positioned in the literature?
* Please point out the evidence for the claim quoted in the third weak point.

---

> ### Author Response · Authors · 2023-11-17
> **Response to Reviewer eFdv (Part I)**
>
> We thank the reviewer for their comments. We have addressed the comments in the rebuttal below.
>
> **Q1:** How the information abundance for the setting of ME is computed.
>
> We acknowledge the reviewer's concern regarding the computation of information abundance under the ME settings in our paper. In our initial version, we did not provide sufficient clarity on this aspect, which may have led to confusion.
>
> To address this issue, we have revised our paper and would like to provide a more accurate and detailed explanation of how the information abundance is computed in the ME setting. Specifically, under the ME settings, we *concatenate the matrices of all embedding sets along the embedding size dimension*. For instance, in the case of calculating the information abundance with a 10x size for ME, as shown in Fig.9 (a), we concatenate all the 10 matrices, each with an embedding size of 10, and then calculate the information abundance of the resulting concatenated matrix. This approach ensures that the information abundance of both the ME and SE (Single Embedding) settings is calculated *under exactly the same matrix shape*, enabling a **fair** comparison between the two.
>
> **Q2:** Related works of multi-facet embedding and polysemy embedding.
>
> We appreciate the reviewer for bringing up the works related to multi-facet embedding and polysemy embedding. We have taken their suggestions into account and incorporated these works into our paper. It is important to noted that the ME design is **only** a small part of our work to reflect how our theory and analysis are applicable to practical scenarios. Our work also makes **significant contributions** to studying the *collapse phenomenon behind the lacked scalability* of recommender models and the *two-sided effect of feature interactions* in recommender models, which are **not covered** by the two works the reviewer provided.
>
> In comparing the ME design with the two works mentioned, we have identified both similarities and differences:
>
> - Similarity: Both ME and Multi-Facet Embedding & Polysemous Embedding introduces *multiple embedding sets* to improve the performance. Multi-Facet Embedding also aims for *better dataset utilization*.
> - Difference (ME vs. Multi-Facet Embedding): While Multi-Facet Embedding utilizes graph decomposition as **prior** knowledge and **explicitly** models the node-aware transition probability to obtain various embedding representations, the ME design focuses more on the collapse phenomenon and works **generally** without prior knowledge or explicit variation modeling.
> - Polysemy Embedding is **specifically** designed for factorization machines (FM) and aims to introduce more non-linearity into FM models. On the other hand, the ME design is applicable to general recommendation models and focuses on improving **dataset utilization**.

---

> ### Author Response · Authors · 2023-11-17
> **Response to Reviewer eFdv (Part II)**
>
> **Q3:** How Finding 1 is believed to applicable to general recommendation models.
>
> We appreciate the reviewer's question and would like to clarify the applicability of Finding 1 to general recommendation models.
>
> Firstly, we acknowledge that our earlier statements may have caused some misunderstanding. It is important to note that merely analyzing the raw embedding matrices is **insufficient** for determining the causes of embedding collapse in general recommendation models, particularly when considering feature interaction. The learning of embeddings is a complex process that involves interactions with **all** other fields, making it challenging to isolate the specific mechanisms responsible for embedding collapse or to evaluate the impact of **field-pair-level** interaction on embedding learning, as discussed in our paper.
>
> To overcome this challenge, we propose a comprehensive analysis from two perspectives, which leads to the formulation of the high-level interaction-collapse law:
>
> 1. Empirical analysis: We conduct empirical analysis *specifically on sub-embedding-based models*. Sub-embeddings serve as bridges to identify field-pairwise influence. It is important to note that the concept of sub-embedding is simply an auxiliary tool for our analysis, and the conclusions drawn from this analysis are not necessarily limited to sub-embedding-based models alone.
>
> 2. Theoretical analysis: We also perform a theoretical analysis focusing on FM-based interaction *without explicit sub-embeddings*, where the embedding matrix interacts directly with all other fields. This theoretical analysis demonstrates that the empirical finding mentioned above is not confined to sub-embedding-based models.
>
> By combining the empirical analysis of sub-embedding-based models and the theoretical analysis of FM-based interaction, we arrive at Finding 1. We argue that this finding should be applicable to **general** recommendation models, as it provides insights into the phenomenon of embedding collapse and its underlying mechanisms.
>
> We hope this clarification addresses the reviewer's concerns and highlights the broader applicability of our findings to the field of recommendation models.
>
> **Q4:** Whether Finding 1 should be a "law" or a "theory".
>
> We appreciate the reviewer's question regarding the classification of our finding as a "law" or a "theory." After careful consideration, we **agree** with the reviewer's viewpoint that the term "law" is typically applied to principles that are derived from extensive experimentation and have a strong empirical basis. In contrast, our Finding 1 is a pattern that has been derived through rigorous analysis and theoretical considerations. Therefore, we agree that it would be more accurate and appropriate to classify it as a "theory" rather than a "law".

---

> ### Author Response · Authors · 2023-11-22
> **Request of Reviewer's attention and feedback**
>
> Dear Reviewer,
>
> This is a kind reminder that the 12 days reviewer-author discussion period only left **less than 1 day**. Please let us know if our response has addressed your concerns.
>
> Following your suggestion, we have answered your concerns and improved the paper in the following aspects:
> - We have **clarified how the information abundance of ME** is computed. Specifically, we compute them by concatenating all embedding sets along the embedding dimension.
> - We have **compared our paper and Multi-Facet Embedding & Polysemous Embedding** to address the contribution and novelty of our work. Our paper contributes distinctively on the embedding collapse issue and the two-sided effect of feature interaction, and ME also differs from these work in some aspects.
> - We have **clarified and rearranged the logic of how to deduce Finding 1** and further **discussed how Finding 1 is applicible to general models**, including how empirical analysis on specific sub-embedding-based models and theoretical analysis on general FM-based models contributes to the deduction.
> - We have **revised the paper and modified *law* into *theory*.**
>
> Thanks again for your valuable review. We are looking forward to your reply and are happy to answer any future questions.

---

> ### Comment · Reviewer_eFdv · 2023-12-04
> **Thanks for the detailed response, my questions are addressed.**
>
> I increased my rating from 5 to 6. Though I would not to champion the paper.

---

### Official Review · Reviewer_DK5H · 2023-11-01

**Soundness:** 3 good
**Presentation:** 3 good
**Contribution:** 3 good
**Rating:** 5
**Confidence:** 4

**Summary:**

The paper addresses challenges in scaling up recommendation models, identifying a phenomenon called "embedding collapse" while enlarging models. The authors introduce information abundance as a metric to measure and evaluate collapse. They show that embedding matrix reside mostly in a low-dimensional subspace in scaled up models. The study analyzes feature interactions in models, noting they reduce overfitting but can also exacerbate embedding collapse. To tackle this, the authors introduce a multi-embedding design, scaling independent embedding sets and integrating specific interaction modules. This approach claims to improve scalability across various recommendation models. Main contributions:
- Highlight non-scalability in recommendation models and define the "embedding collapse" phenomenon.
- Empirical and theoretical analysis reveals the dual impact of feature interaction on scalability.
- Introduction of the multi-embedding design to achieve scalability improvements for recommendation models.

**Strengths:**

Originality:
- ‘Information Abundance' as a quantitative novel measure to measure the embedding layer collapse.
- The 'Interaction-Collapse Law' and the two sided effect of feature interaction process helps improve the understanding of embeddings' behavior in recommendation systems.

Quality:
- The authors have detailed exploration of embeddings and their behavior, particularly in the context of information collapse with rigorous visualizations.

Significance:
- Broad Implications for Recommendation Systems: Given the ubiquitous nature of recommendation systems in today's digital platforms, insights into their workings, particularly regarding embeddings, have widespread implications.
- Potential for Future Research: Introducing novel concepts and metrics invariably opens the door for future studies, both to validate and to build upon these ideas. The 'Interaction-Collapse Law', for instance, may become a focal point in subsequent research.

**Weaknesses:**

1. Insufficient Empirical Validation on Large-Scale Data: The authors have shown with empirical evidences that large scale recommendation models scale poorly. However it is a common knowledge that large scale models are inherently data hungry to achieve better model convergence. This is an important premise that the paper relies on, it would good if authors can follow up to prove/disprove this as additional data points in this paper. The experiments seem to be on same amount of training data on scaled up models which does not unlock the full power of the scaled up model

2. Studies on Collapse and its effect on scalability and overfitting seem to be limited to Single Embedding studies. To make a stronger case about the proposed method, it would be great if authors can provide discussion on the same for proposed Multi Embedding Design.

**Questions:**

1. How do you justify the experiment setup when we know that bigger models are inherently data hungry and need more training data to achieve full convergence?

2. From the experiment results shown, the single embedding and multi embedding cases show marginal improvement in AUC values. Can authors provide more context on the significance of the improvements here and if they are well outside the noise region?

---

> ### Author Response · Authors · 2023-11-17
> **Response to Reviewer DK5H (Part I)**
>
> We thank the reviewer for their comments. We have addressed the comments in the rebuttal below.
>
> **Q1:** Scaling law under different data size.
>
> We sincerely appreciate the reviewer's suggestion to evaluate the scalability of our model on larger datasets. However, it is important to note that the Criteo and Avazu datasets, which we used for our experiments, are currently among the **largest publicly available benchmark datasets**, containing approximately $4\times 10^8$ instances. Unfortunately, there are **no** larger datasets available for us to incorporate into our analysis. Nevertheless, we are confident that our proposed approach can be effectively applied to real-world industrial scenarios where significantly larger amounts of data are available.
>
> Moreover, we understand the reviewer's concern about the scalability of recommendation models and the need to examine their performance under different data sizes. To address this concern, we performed additional experiments by *downsampling the original Criteo dataset to 10% and 50% of its size*. We evaluated both the base model and the 4x model on these reduced datasets, and the results are presented in the following table:
>
> | DCNv2 |    1x   |    4x   |
> |:-----:|:-------:|:-------:|
> |  10%  | 0.79494 | 0.79560 |
> |  50%  | 0.80892 | 0.80920 |
> |  100% | 0.81339 | 0.81372 |
>
> As shown in the table, even when the amount of data is significantly reduced, the 4x model consistently outperforms the base model. These results clearly demonstrate that scaling up the model size leads to performance improvements, **even with smaller datasets**. Therefore, we firmly believe that our approach is scalable and capable of delivering enhanced recommendation performance in various data scenarios. We hope this clarification adequately addresses the reviewer's concern and reinforces the validity and scalability of our proposed algorithm.
>
> **Q2:** Study of collapse and overfitting for ME.
>
> We appreciate the reviewer's question regarding the study of collapse and overfitting for ME (Multi-Embedding) models. In response, we have made significant revisions in Appendix to the paper and *conducted additional experiments* to address this concern. Allow us to present the updated findings:
>
> 1. Evidence II on ME: To further investigate the feature interactions of ME, we have *calculated the information abundance of sub-embeddings in an ME DCNv2 model*. These sub-embeddings are obtained by concatenating the projected embeddings from each embedding set. Specifically, we define sub-embedding $E_i^{\to j}$ as $\left[E_i^{(1)}(W_{i,j}^{(1)})^{\top}, ..., E_i^{(m)}(W_{i,j}^{(m)})^{\top}\right]$. We visualize these sub-embeddings in Fig. 14. Notably, we observe that the influence of the field $j$ on $\mathrm{IA}(E_i^{\to j})$ is significantly reduced in the ME model compared to the SE (Single-Embedding) model. This finding suggests that *ME mitigates the collapse caused by feature interaction*.
>
> 2. Evidence III on ME: In order to further investigate the impact of ME on collapse and overfitting, we conducted *regularization experiments* on the ME DCNv2 model. The results are summarized in the table below and Fig. 15:
>
> |            DCNv2 |    1x   |    2x   |    3x   |    4x   |   10x   |
> |-----------------:|:-------:|:-------:|:-------:|:-------:|:-------:|
> | SE (regularized) | 0.81313 | 0.81318 | 0.81311 | 0.81299 | 0.81307 |
> |    SE (standard) | 0.81339 | 0.81341 | 0.81345 | 0.81346 | 0.81357 |
> | ME (regularized, new) |         | 0.81324 | 0.81343 | 0.81349 | 0.81364 |
> |    ME (standard) |         | 0.81348 | 0.81361 | 0.81372 | 0.81385 |
>
> From the table, we observe that the regularized ME DCNv2 model exhibits a similar increasing trend in performance with respect to the embedding size as the regularized SE model. However, the performance decline in the regularized ME model is less significant compared to the regularized SE model. This finding provides further evidence that ME provides model scalability even when applyed on an interaction layer to cause overfitting.
>
> In conclusion, the evidence from our experiments supports the effectiveness of ME in mitigating collapse caused by feature interaction.

---

> ### Author Response · Authors · 2023-11-17
> **Response to Reviewer DK5H (Part II)**
>
> **Q3:** Significance of performance improvement.
>
> In response to the reviewer's question regarding the significance of our performance improvement, we would like to highlight several key points. Firstly, we acknowledge that previous works have regarded a gain of **1e-3** as already significant [1-3]. However, our ME design, while being relatively simple, is comparable with this threshold by achieving a performance gain of 7e-4 and 1.4e-3 on the Criteo and Avazu datasets respectively, solely through the scaling up of the model size. This achievement, which was **not** accomplished by the original SE, demonstrates the remarkable nature of our approach.
>
> Furthermore, we would like to draw attention to the standard deviation of the performance, as presented in Table 1 of our paper (Appendix C.3). Our improvements **surpass** the magnitude of the standard deviation, indicating that they are **not** merely a result of random variation. To further validate the significance of our findings, we conducted a t-test and obtained a p-value of 1.6e-3 for DCNv2 on the Criteo dataset (similar results were observed for other experiments). This p-value confirms that the observed performance improvement is not attributable to randomness, but rather represents a substantial and meaningful advancement.
>
>
> [1] Deepfm: a factorization-machine based neural network for ctr prediction. In IJCAI, 2017.
> [2] Autoint: Automatic feature interaction learning via self-attentive neural networks. In CIKM, 2019.
> [3] DCN V2: Improved Deep & Cross Network and Practical Lessons for Web-scale Learning to Rank Systems. In WWW, 2021.

---

> ### Author Response · Authors · 2023-11-22
> **Request of Reviewer's attention and feedback**
>
> Dear Reviewer,
>
> This is a kind reminder that the 12 days reviewer-author discussion period only left **less than 1 day**. Please let us know if our response has addressed your concerns.
>
> Following your suggestion, we have answered your concerns and improved the paper in the following aspects:
> - We have **discussed the data size issue in recommender systems**, and show with **supplemental experiments** that, even with small data size, the existing embedding size is still not enough.
> - We have **added analysis study on ME models** as those in Sec 4. The results show that ME can properly mitigate collapse and improve scalability.
> - We have **highlighted the significance of performance improvement**, including how the improvement is not marginal and how our improvement is a breakthrough, and further discussed how the improvement surpasses the scope of variance.
>
> Thanks again for your valuable review. We are looking forward to your reply and are happy to answer any future questions.

---

### Official Review · Reviewer_JdNi · 2023-11-01

**Soundness:** 3 good
**Presentation:** 2 fair
**Contribution:** 1 poor
**Rating:** 5
**Confidence:** 3

**Summary:**

This paper suggests that the embedding collapse phenomenon restricts the scalability of existing recommendation models. Empirical and theoretical analysis show that interaction with collapsed embeddings constrains embedding learning. Also, this paper proposes a multi-embedding design incorporating embedding-set-specific interaction modules to capture diverse patterns and reduce collapse.

**Strengths:**

S1. This paper provides empirical and theoretical analysis of the embedding collapse phenomenon.

S2. This paper provides information abundance for quantifying the degree of collapse for such matrices with low-rank tendencies.

**Weaknesses:**

W1. The novelty of this paper seems to be limited. The method of dividing the single embedding into multi-embedding sets is similar to DMRL[1] for disentangled representation learning. DMRL divides the feature representation of each modality into k chunks. As a result, the features of different factors are entangled.

W2. The motivation is not completely solid. The reason for increasing the embedding size of the model is inappropriate.

W3. The experimental results of the paper are insufficient. When the embedding size was scaled up through multi-embedding, the experimental results show that performance increases. However, the performance improvement is marginal.

[1] Disentangled Multimodal Representation Learning for Recommendation, IEEE’22

**Questions:**

Depending on the model and dataset, the model will have an appropriate embedding size. Therefore, it seems reasonable that the performance will drop if the embedding size deviates from the proper value. Then. do we need to scale up the embedding size for the same dataset?

---

> ### Author Response · Authors · 2023-11-17
> **Response to Reviewer JdNi (Part I)**
>
> We thank the reviewer for their comments. We have addressed the comments in the rebuttal below.
>
> **Q1:** Relation between multi-embedding and DMRL.
>
> We appreciate the reviewer's question regarding the relationship between ME and DMRL, and we have incorporated it into our paper. It is important to note that the ME design is **just one aspect** of our overall work that **reflects** how our theory and analysis apply to practical scenarios. Our work also makes **significant contributions** to studying the collapse phenomenon behind the lack of scalability of recommender models and the two-sided effect of feature interactions in recommender models, which are never covered in DRML.
>
> In comparing the ME design and DMRL, we identify the following similarities and differences:
> - Similarity: Both ME and DMRL propose the idea of introducing *multiple embedding layers*.
> - Differences: (1) DMRL is a **specific** method, whereas ME is a **general** framework that can be applied to various feature interaction designs. (2) While DMRL focuses on learning different factors and **explicitly "decoupling"** them, ME takes a collapse perspective and demonstrates that even without decoupling, ME performs better than Single Embedding (SE) as it mitigates the collapse, a point that DMRL does **not** explore.
>
> **Q2:** Reason for increasing the embedding size of the model.
>
> This question can be divided into two parts: why the embedding size should be increased instead of the size of other modules (e.g., the number of layers in the cross layer or the number of layers in the MLP), and why the existing embedding size is not large enough, which can be summarized as *"why embedding" and "why increase"*.
>
> Regarding "why embedding", it is widely recognized in both the academic and industrial communities that the embedding layer accounts for the **largest** number of parameters (>92% in our DCNv2 baseline for Criteo, and even higher for industrial models), making it a crucial and informative **bottleneck **in the model. To further illustrate this point, we conducted experiments to *increase the number of interaction layers and MLP layers* in the DCNv2 baseline and recorded the results in the following table:
>
> |                 DCNv2 |    1x   |    2x   |    4x   |
> |----------------------:|:-------:|:-------:|:-------:|
> |              standard | 0.81339 | 0.81341 | 0.81346 |
> | *+ #cross layer* | 0.81325 | 0.81338 | 0.81344 |
> |   *+ #MLP depth* | 0.81337 | 0.81345 | 0.81342 |
>
> From the table, it can be observed that increasing the number of cross layers or MLP layers does **not** lead to improved performance. Therefore, it is reasonable and necessary to scale up the embedding size.
>
> As for "why increase," the motivation lies in the fact that existing models have **too small** an embedding size that is inadequate for capturing and preserving data information effectively. For instance, the Johnson-Lindenstrauss lemma suggests that a projection over a $d$-dimensional space needs to have a dimension of approximately $\Omega\left(\frac{\log d}{\epsilon^2}\right)$ to preserve pairwise distances within $1\pm \epsilon$. However, the existing embedding size of 10 does not match the maximum field cardinality, which can be on the order of $10^5$. Our experimental results using ME further validate that increasing the embedding size in a proper manner is promising to lead to performance improvements, indicating that the existing embedding size is indeed insufficient.
>
> Furthermore, it is important to emphasize that our work is **not** solely focused on enhancing model performance at the same small size. Instead, our goal is to **overcome** the size limitations imposed on recommendation models by existing architectures and achieve consistent positive performance improvements when scaling up the models (as stated in the Introduction section). ME is a simple yet effective design that can be applied to general architectures and yield performance gains. We anticipate that ME can be combined with future works that aim to improve performance through other means.

---

> ### Author Response · Authors · 2023-11-17
> **Response to Reviewer JdNi (Part II)**
>
> **Q3:** Significance of experiment results.
>
> Numerous existing studies have indicated that even a gain as small as **1e-3** is considered significant [1-3]. In our ME design, despite its simplicity, we were able to achieve a performance gain of 7e-4 on the Criteo dataset and 1.4e-3 on the Avazu dataset by **solely scaling up the model size**. Notably, this level of improvement was **never** achieved by the original SE approach. We firmly believe that such remarkable enhancement demonstrates the effectiveness and potential of our proposed algorithm.
>
> [1] Deepfm: a factorization-machine based neural network for ctr prediction. In IJCAI, 2017.
> [2] Autoint: Automatic feature interaction learning via self-attentive neural networks. In CIKM, 2019.
> [3] DCN V2: Improved Deep & Cross Network and Practical Lessons for Web-scale Learning to Rank Systems. In WWW, 2021.
>
> **Q4:** Appropriate embedding sizes.
>
> We appreciate the reviewer's question regarding the "appropriate embedding size". In our paper, as demonstrated in Table 1, we acknowledge that the determination of the appropriate embedding size (*underlined* or **bolded**) relies on *both the dataset and the model itself*. Interestingly, our findings indicate that ME models generally require larger embedding sizes compared to SE models.
>
> However, it is important to note that our objective is **not** to indiscriminately increase the model size *using any fixed (possibly SE) model architecture*. While SE models may typically have smaller appropriate embedding sizes, it does **not** necessarily mean that recommendation models employing such embedding sizes effectively exploit the dataset's information. This discrepancy arises from inherent limitations in the model design, as we have identified through our analysis.
>
> Also due to this limitation of existing (SE) models, our approach focuses on: firstly proposing a *more scalable* model design, and secondly *scaling up* the newly introduced scalable model. Our ME model serves as a straightforward and convenient scalable design, exhibiting a *larger appropriate embedding size* in conjunction with improved performance.

---

> ### Author Response · Authors · 2023-11-22
> **Request of Reviewer's attention and feedback**
>
> Dear Reviewer,
>
> This is a kind reminder that the 12 days reviewer-author discussion period only left **less than 1 day**. Please let us know if our response has addressed your concerns.
>
> Following your suggestion, we have answered your concerns and improved the paper in the following aspects:
> - We have **compared our paper and DMRL** to address the contribution and novelty of our work. Our paper contributes distinctively on the embedding collapse issue and the two-sided effect of feature interaction, and ME also differs from DRML in some aspects.
> - We have **clarified the reason for increasing the embedding size**, including how the embedding serves as a bottleneck and how the existing embedding size is too small.
> - We have **highlighted the significance of performance improvement**, including how the improvement is not marginal and how our improvement is a breakthrough.
> - We have **discussed the concept of "appropriate embedding size"** and show how our ME is releated with the concept of appropriate embedding size under the perspective of scaling up models.
>
> Thanks again for your valuable review. We are looking forward to your reply and are happy to answer any future questions.

---

### Official Review · Reviewer_9BGA · 2023-11-01

**Soundness:** 2 fair
**Presentation:** 3 good
**Contribution:** 2 fair
**Rating:** 5
**Confidence:** 4

**Summary:**

This paper studies recommendation model performance when scaling up the embedding layers of the model. The paper identifies a phenomenon of embedding collapse, wherein the embedding matrix tends to reside in a low-dimensional subspace. Through empirical experiments on FFM and DCNv2 and theoretical analysis on FM, the paper shows that the feature interaction process of recommendation models leads to embedding collapse and thus limits the model scalability. The paper also performed empirical experiments on regularized DCNv2 and DNN which led to less collapsed embeddings, but the model performance got worse. The paper proposes multi-embedding, which leads to better performance when scaling up the embedding layers. Experiments demonstrate that this proposed design provides consistent scalability for various recommendation models.

**Strengths:**

Originality:
- The paper investigates the enlarged embedding layers of recommendation models and identifies a phenomenon of embedding collapse, wherein the embedding matrix tends to reside in a low-dimensional subspace. The discovery is novel as far as I know.
- The paper proposed information abundance to measure the degree of collapse for embedding matrices.

Quality:
- The paper is well-written. It starts with a novel finding of embedding collapse when increasing embedding dimension which might lead to poor scalability, performs empirical experiments and theoretical analysis to show that it is caused by feature interaction, and proposes a solution to increase scalability.

Significance:
- The scaling law of recommendation models is an important topic in both academia and industry. This paper investigates the scaling law of embedding layers and shows why naively increasing embedding dim is not sufficient for scalability. The paper proposed multi-embedding which could help alleviate the phenomenon of embedding collapse and improve scalability.

**Weaknesses:**

- In section 3, the paper proposes Information Abundance to measure the degree of collapse of embedding matrices. As the paper focuses on the scaling law of embedding layers, the paper should discuss whether Information Abundance is a fair metric when comparing embedding matrices of different dimension sizes.
- In section 4.2, the paper uses regularized DCNv2 as an example to show that suppressing feature interaction is insufficient for scalability. It is unclear to me why feature interaction in regularized DCNv2 is suppressed.
- The performance of multi-embedding lacks ablation study. One example is that in Figure 7 (right), we can feed all 4 small embeddings into a single feature interaction layer, and test its performance against the proposed approach. We can also test the Information Abundance of such design versus the proposed multi-embedding design.
- It would be interesting to test the scaling law of embedding layers with increased feature interaction complexity. This can help us better understand whether the embedding collapse phenomenon is caused by insufficient feature interaction complexity.

**Questions:**

My questions are listed in the weaknesses section.

---

> ### Author Response · Authors · 2023-11-17
> **Response to Reviewer 9BGA (Part I)**
>
> We thank the reviewer for their comments. We have addressed the comments in the rebuttal below.
>
> **Q1:** How the information abundance is a fair metric.
>
> We appreciate the reviewer's question regarding the fairness of information abundance as a metric. Allow us to address this concern and clarify our usage of information abundance throughout the paper.
>
> Firstly, the embedding size influences information abundance. To ensure fair comparisons and mitigate this influence, we specifically compare the information abundance of matrices **only** with the same embedding size throughout this paper. We have listed *all the instances* where we utilize information abundance in this manner:
>
> - Fig.2: We compare the learned embedding matrix with a randomly initialized one, both having an *embedding size of 100*.
> - Fig.3: We compare the information abundance of two sub-embedding matrices, denoted as $E_i^{\to j_1}$ and $E_i^{\to j_2}$, which have the same *sub-embedding size of 10*.
> - Fig.4: We compare the information abundance of embedding matrices with the same shape but varying field configurations. *The embedding size in this case is 5*.
> - Fig.5 & 6: We compare the regularized DCNv2 and DNN models with the standard DCNv2 model. All models considered in these figures have the same *embedding size of 100*.
> - Fig.9 (a): We compare two types of embedding methods, SE and ME, with an embedding size scaled by 10x. *The embedding size in this case is 100.*
> - Fig.9 (b): We compare four sets of embeddings, each with the same *embedding size of 10*.
>
> Furthermore, it is essential to note that information abundance serves as a **simple and convenient** metric for assessing the degree of collapse in our work. The purpose of information abundance in our work is to provide a quantifiable measure that enables meaningful analysis. While it is not explicitly adapted to different embedding sizes in this work, we acknowledge the potential for such extensions. For example, possible extensions could include normalizing information abundance by the embedding size, such as using $\frac{\mathrm{IA}(E)}{K}$ or $\frac{\mathrm{IA}(E)}{\mathbb{E}[\mathrm{IA}(\text{randn}(E))]}$, where $K$ represents the embedding size. We *add in Fig. 12* as the comparision with different embedding sizes through the former approach, and observed that the *normalized information abundance decreases with the embedding size*, which is consistent with the observation in Fig.1 (b).
>
> **Q2:** How feature interaction is surpressed in regularized DCNv2.
>
> We appreciate the reviewer's question regarding the suppression of feature interactions in regularized DCNv2. We recognize the confusion caused by our previous description and would like to clarify and provide a more accurate explanation.
>
> In our study, we investigated the issue of embedding collapse from the perspective of feature interactions. To address this problem, we made two modifications to DCNv2, a well-designed explicit-interaction model: (1) revising the modules in DCNv2 that **contribute to collapse** (referred to as Evidence III of Regularized DCNv2) and (2) directly avoiding explicit feature crossings (referred to as Evidence IV of DNN).
>
> Regarding Evidence II, we recognize that $W_i^{\to j}$ can **lead to collapse** in sub-embeddings by projecting embeddings into a more collapsed space. By regularizing $W_i^{\to j}$ to be a unitary matrix (or the multiplication of unitary matrices), we ensure the **preservation** of all singular values of the sub-embedding. Consequently, the information abundance of sub-embeddings in regularized DCNv2 is larger than standard DCNv2. We provide a *supplementary figure, Fig. 13*, illustrating the information abundance of sub-embeddings in standard and regularized DCNv2. This figure clearly demonstrates that sub-embeddings in regularized DCNv2 *exhibits a higher information abundance*. Based on our Finding 1, regularized DCNv2 mitigates the problem of embedding collapse since sub-embeddings are *directly interacted with the embeddings*.
>
> We hope this clarification addresses the concerns raised by the reviewer regarding the suppression of feature interactions in regularized DCNv2.

---

> ### Author Response · Authors · 2023-11-17
> **Response to Reviewer 9BGA (Part II)**
>
> **Q3:** Ablation study of multi-embedding.
>
> Thank you for your question regarding the ablation study of multi-embedding. In our initial submission, we presented an ablation experiment in Fig. 9 (d) where the feature interaction layers were **not strictly shared but implicitly weight-aligned**. However, we acknowledge that we should have included the reviewer's proposed *most straightforward ablation experiment*. We have addressed this oversight by adding the suggested experiment and present the results in the latest revision, as shown in the table below:
>
> |                              DCNv2 |    1x   |    2x   |    3x   |    4x   |   10x   |
> |-----------------------------------:|:-------:|:-------:|:-------:|:-------:|:-------:|
> |                                 SE | 0.81339 | 0.81341 | 0.81345 | 0.81346 | 0.81357 |
> | ME, independent interaction (ours) |         | 0.81348 | 0.81361 | 0.81372 | 0.81385 |
> | ME, weight norm aligned (ablation) |         | 0.81332 | 0.81340 | 0.81334 | 0.81337 |
> | *ME, shared interaction (ablation, new)* |         | 0.81302 | 0.81283 | 0.81291 | 0.81221 |
>
> From the table, it is evident that the multi-embedding (ME) with shared interaction performs *worse than ME with specific interactions* (and even worse than the single embedding, SE). This finding is consistent with our analysis that ME *works from the diversity of interaction layers*.
>
> **Q4:** Scaling law with increased feature interaction complexity.
>
> In addressing the impact of feature interaction complexity on scaling laws, we bifurcate the complexity into two distinct components: *interaction architectures* and *the number of interaction parameters*.
>
> Regarding interaction architectures, even when we consistently employed the DCNv2 architecture, **renowned for its complexity**, the results did not exhibit substantial performance enhancement. This finding suggests that the complexity of the interaction architecture, while important, may **not** be the primary driver in the scaling laws of recommendation models.
>
> For the second component – the number of interaction parameters – we further augment our analysis. We *increased the number of feature interaction layer parameters* with the scaling of embedding sizes. The table below encapsulates our findings:
>
> |                 DCNv2 |    1x   |    2x   |    4x   |
> |----------------------:|:-------:|:-------:|:-------:|
> |              standard | 0.81339 | 0.81341 | 0.81346 |
> | *+ #cross layer* | 0.81325 | 0.81338 | 0.81344 |
>
> This table reveals that increasing the number of cross layers does **not** proportionally enhance performance even in scenarios of increased embedding sizes, indicating that merely increasing the number of interaction parameters does not necessarily correspond with an improvement in model performance.

---

> ### Author Response · Authors · 2023-11-22
> **Request of Reviewer's attention and feedback**
>
> Dear Reviewer,
>
> This is a kind reminder that the 12 days reviewer-author discussion period only left **less than 1 day**. Please let us know if our response has addressed your concerns.
>
> Following your suggestion, we have answered your concerns and improved the paper in the following aspects:
> - We have **clarified how information abundance can be properly applied**, including details of usage in our experiments and possible extensions for cases with different embedding sizes.
> - We have **clarified how regularized DCNv2 can mitigate collapse** by surpressing the mechenism in feature interaction that leads to collapse, and supplement detailed results in the revised paper.
> - We have **added new ablation study** that all embeddings sets shares one feature interaction module, and results show that the utilization of embedding-set-specific feature interaction is crucial.
> - We have **add analysis experiments of scaling up with increased feature interaction complexity**, and show that improving feature interaction complexity does not lead to performance gain, inferring that the bottleneck of performance is the embedding size.
>
> Thanks again for your valuable review. We are looking forward to your reply and are happy to answer any future questions.

---

### Meta-Review · Area_Chair_WmSF · 2023-12-07

**Metareview:**

The paper identifies an issue with recommender systems, where models with a large embedding size suffer from embedding collapse. The authors first define the problem and demonstrate when it occurs, then offer a solution for it via a multi-embedding design. The reviews have an overall positive feedback on the first part. They mention that this phenomenon is important to highlight, and that the experiments demonstrating it are convincing in showing that indeed this collapse occurs. The concerns are mostly directed towards the second part, meaning the proposed solution. The particular weaknesses mentioned vary among reviews, but all of the reviews mention it to be unconvincing in some way. The authors provided a response that mitigated some of these concerns, but to me the scope of the work required in order to fully fix the highlighted issues seem too large, and it is likely that without another round of reviews, this second part (the solution to the problem) will remain problematic. Since this is a significant part of the paper I tend to recommend rejecting the paper in its current form.

**Justification For Why Not Higher Score:**

The proposed solution to the raised problem is not convincing enough. It requires a more thorough analysis

**Justification For Why Not Lower Score:**

n/a

---

### Decision · Program_Chairs · 2024-01-16

Reject